# Bone Morphogenic Proteins in Pediatric Diffuse Midline Gliomas: How to Make New Out of Old?

**DOI:** 10.3390/ijms25063361

**Published:** 2024-03-15

**Authors:** Clément Berthelot, Paul Huchedé, Adrien Bertrand-Chapel, Pierre-Aurélien Beuriat, Pierre Leblond, Marie Castets

**Affiliations:** 1Childhood Cancer & Cell Death Team (C3 Team), LabEx DEVweCAN, Institut Convergence Plascan, Centre de Recherche en Cancérologie de Lyon (CRCL), Centre Léon Bérard, Université Claude Bernard Lyon 1, INSERM 1052, CNRS 5286, 69008 Lyon, France; clement.berthelot@lyon.unicancer.fr (C.B.); paul.huchede@lyon.unicancer.fr (P.H.); adrien.bertrand@lyon.unicancer.fr (A.B.-C.); pierre.leblond@ihope.fr (P.L.); marie.castets@lyon.unicancer.fr (M.C.); 2South-ROCK Pediatric Cancer Research Center, 69008 Lyon, France; 3Multisite Institute of Pathology, Groupement Hospitalier Est du CHU de Lyon, Hopital Femme-Mère-Enfant, 69677 Bron, France; 4Department of Translational Research in Pediatric Oncology PROSPECT, Centre Léon Bérard, 69008 Lyon, France; 5Department of Pediatric Oncology, Institut d’Hématologie et d’Oncologie Pédiatrique, Centre Léon Bérard, 69008 Lyon, France

**Keywords:** DIPG, DMG, BMP, H3K27M, pediatric glioma, quiescence, invasion

## Abstract

The BMP pathway is one of the major signaling pathways in embryonic development, ontogeny and homeostasis, identified many years ago by pioneers in developmental biology. Evidence of the deregulation of its activity has also emerged in many cancers, with complex and sometimes opposing effects. Recently, its role has been suspected in Diffuse Midline Gliomas (DMG), among which Diffuse Intrinsic Pontine Gliomas (DIPG) are one of the most complex challenges in pediatric oncology. Genomic sequencing has led to understanding part of their molecular etiology, with the identification of histone H3 mutations in a large proportion of patients. The epigenetic remodeling associated with these genetic alterations has also been precisely described, creating a permissive context for oncogenic transcriptional program activation. This review aims to describe the new findings about the involvement of BMP pathway activation in these tumors, placing their appearance in a developmental context. Targeting the oncogenic synergy resulting from this pathway activation in an H3K27M context could offer new therapeutic perspectives based on targeting treatment-resistant cell states.

## 1. Introduction

Diffuse Intrinsic Pontine Gliomas (DIPG), which belong to the Pediatric Diffuse Midline Gliomas (pDMG) group, are one of the most devastating cancers in children [1,2]. These rare and aggressive brain tumors arise exclusively in the midline structures of the brain, notably in the pons for the so-called DIPG, complicating their therapeutic management by surgical resection [3,4]. Despite years of clinical trials, they are almost uniformly fatal, representing the leading cause of mortality in pediatric neuro-oncology [5,6]. The median overall survival is 12 months and varies according to tumor location and underlying genetic etiology [3]. One of the determinants of tumors’ aggressiveness is their highly infiltrative nature. In DIPG, the tumor is usually centered within the pons at diagnosis, but disease progression is frequently associated with leptomeningeal infiltration and, in some cases, with metastases spreading in distant regions of the central nervous system [7,8]. At present, radiotherapy remains the treatment of reference in clinical practice, although its beneficial effect is only transient. Identifying tumor vulnerabilities is, therefore, crucial to develop new therapeutic approaches.

A breakthrough in our understanding of the molecular basis of pDMG has been achieved with the identification of K27M mutations in histone H3.1 or H3.3 in almost 80% of cases ([3,9]; Table 1). These mutations, as well as the more recently identified overexpression of the EZH2 inhibitor protein EZHIP, all lead to massive epigenetic reprogramming, which underlies disease etiology [4]. Although these abnormalities result in chromatin decondensation, key differences in chromatin states exist between H3.1K27M and H3.3K27M, which then display distinct modes of oncogenic reprogramming [10]. These molecular differences likely explain a first degree of patients’ heterogeneity characterized by differences in clinical outcomes [11,12,13]. In addition to this inter-tumor heterogeneity, these cancers also display genetic [14,15] and epigenetic intra-tumor heterogeneity. The latter was recently identified as a result of the specific pattern expression of each mutant histone, resulting in distinct epigenetic cell states [16]. In turn, this epigenetic intra-tumor heterogeneity must be integrated with the different transcriptomic cell states identified so far [9,17]. More generally, understanding how the complex interplay between epigenomic rewiring caused by H3 mutations and oncogenic transcriptional signaling pathways govern the evolutionary dynamics of different cell states under the pressure of tumor–autonomous or microenvironment-driven factors is crucial to optimize therapeutic intervention. One of the challenges is to understand how oncogenic nodes could appear under the concomitant effect of this altered H3.3K27 epigenetic context and the activation of transcriptional pathways, particularly developmental ones. Indeed, latest studies have highlighted the fact that the epigenetic remodeling induced by H3K27M mutations is not sufficient to induce tumor transformation but rather creates a permissive context for the activation of oncogenic reprogramming, notably induced by transcriptional cascades [18,19,20].

Here, we report and discuss recent evidence about the role of the Bone Morphogenic Protein (BMP)-induced signaling cascades in the oncogenic transcriptomic rewiring, leading to pDMG.

## 2. BMP Family: An Old Hand in Embryonic Development

As their name indicates, BMPs were originally discovered in 1965 by Marshall Urist for their osteoinductive properties [22]. They started to be formally identified only in the 1980s, with the successive purification and/or cloning of *BMP2*, *BMP3* and *BMP4*. Since then, a wide variety of roles has been attributed to BMP signaling, including gastrulation and embryonic development [23], skeleton and limb formation [24], central nervous system (CNS) development [25], as well as homeostatic functions in adult tissues [26,27].

Part of the TGF-β superfamily [28,29], the BMP family now comprises sixteen ligands that exert their effects by binding to seven receptors, leading to a range of possible combinations with varied downstream consequences. Importantly, BMP ligands have both redundant and specific functions. BMP receptors, responsible for signal transduction, are classified in two groups: (i) type I composed of ALK1/ACVRL1, ALK2/ACVR1, ALK3/BMPR1a, and ALK6/BMPR1b, and (ii) type II composed of BMPR2, ACTR-IIA/ACVR2A and ACTR-IIB/ACVR2B. Ligands are, thus, bound to a hetero-tetrameric complex, comprising two distinct type I receptors and two type II receptors [30,31,32], with the added possibility of heterodimeric ligands providing an extra level of control over signal intensity and specificity. Ligands/receptors binding trigger two types of signaling cascades [29]: (i) the so-called canonic pathway, which is mediated by phosphorylation-driven activation of Smad1/5/8, leading to their translocation into the nucleus and the subsequent activation of a transcriptional program [33,34], and (ii) the Smad-independent non-canonical pathway mediated by the activation of the mitogen-activated protein kinase (MAPK) pathways, including p38, ERK1/2 and JNK [29,35]. In all cases, the activation of the pathway is finely regulated by cell-autonomous and non-autonomous mechanisms, and notably via inhibitors, acting extra- and intra-cellularly [36,37,38,39].

These inhibitory mechanisms are involved in the establishment of the dorso/ventral (DV) gradient of BMP, which is essential for the correct patterning of the embryo. Indeed, from their discovery until now, BMPs have emerged as major players in the regulation of cell fate during development through a precise and dynamic spatio-temporal patterning [40,41]. Along the dorsal BMP-low to ventral BMP-high axis, BMPs participate in the definition of the identity of the three germ layers, i.e., ectoderm, mesoderm, and endoderm. When gastrulation starts, low and high BMP contents, respectively, specify dorsal (e.g., somites) versus ventral fates (e.g., lateral lamina) in the mesoderm [40].

Beyond the regulation of mesoderm derivatives’ identity, BMPs play a pivotal role in CNS development and patterning (24, Figure 1). As identified over 100 years ago by Spemann and Mangold, inhibition of the BMP pathway is initially required for neurectoderm differentiation during gastrulation [42]. BMPs’ role extends beyond these early stages of embryogenesis since the activation of the pathway is necessary for the appropriate specification of the neural stem cells’ (NSCs) fate [43,44,45]. NSCs are multipotent progenitors that can differentiate into neurons or glial cells, including astrocytes and oligodendrocytes in the CNS. Under the effect of BMP secretion by the ectoderm and the roof of the neural tube, a dorsal-high/ventral-low gradient establishes the polarity of the neural tube [46,47,48,49,50]. This gradient first promotes neurogenesis during the early stages of development, committing NSCs to a neuronal phenotype. Later, including in the post-natal period, BMP activation switches from this pro-neuronal activity to the promotion of gliomagenesis, and mostly induces astroglial differentiation [51,52]. Indeed, activation of the canonical SMAD1/5/8-dependent pathway leads to an upregulation of the inhibitor of differentiation-1 (ID1) expression, both known to degrade pro-neurogenic factors such as Mash1 [53,54] and to associate with the signal transducer and activator of transcription-3 (STAT3) to promote astrocyte differentiation [55,56]. Concomitantly with this pro-astroglial effect, BMP activation antagonizes oligodendrocyte progenitor cell (OPC) differentiation by inducing ID2/ID4 expression. Indeed, ID2/ID4 prevent the nuclear translocation of Olig1/2 transcription factors that specifically guide oligodendrocyte differentiation by inducing oligogenic gene expression, such as myelin basic protein (MBP) or 2′,3′-cyclic nucleotide 3′-phosphodiesterase (CNP) [57,58,59,60]. BMP signaling also plays a critical role in maintaining adult NSC niches in the subventricular zone (SVZ) and subgranular zone (SGZ) to allow neuronal and glial regeneration, indicating that the role of BMPs in the CNS is not limited to antenatal development.

This global vision of the role of BMPs in CNS patterning/homeostasis needs to be qualified and complexified by the specific features of the ligand/receptor pairs involved in these different regulatory sequences. BMP2, BMP4 and BMP7 have notably been largely involved in the different steps of the CNS patterning and in the determination of NSC fate. Alteration of BMP7 expression was shown to result in embryo dorsalization, with an aberrant induction of neural differentiation in ectoderm [61,62], while treatment with BMP2 was shown to revert a similar phenotype induced by the mutation of its ortholog Dpp (Decapentaplegic) in *Drosophila* [63,64]. Of note, BMP7 specifically influences the patterning and the growth of the developing hindbrain, notably composed of the pons and the cerebellum [65,66], which suggests a spatial specificity among the BMP family. Similarly, the deletion of BMP receptor genes *BMPR1A* and *BMPR1B* in the mouse telencephalon results in the loss of all dorsal midline structures without affecting the specification of cortical and ventral precursors [67,68,69].

While the activity of the BMP pathway is precisely regulated in time and space to ensure the proper development and specification of all tissues, its action is also finely coordinated with that of other morphogens during development, starting sequentially with Fibroblast Growth Factor (FGF) during gastrulation, then WNT and Sonic hedgehog (Shh) during neural tube patterning, and ending with NOTCH during the switch from neurogenesis to gliomagenesis [49,50,70,71,72]. This precise coordination of morphogen-induced transcriptional regulatory cascades is intertwined with epigenetic modifications, which ultimately lead to cell fate determination. For example, the expression of the BMP-inhibitor Noggin, which is induced by BMP activation during dorsal interneuron generation as part as a negative-feedback regulatory loop, depends on the modulation of H3K27me3 levels at the Noggin promoter. This demethylation is mediated by the interaction between histone demethylase KDM6B (JMJD3) and the Smad1/Smad4 complex, adding the regulation of BMP activation levels by inhibitors to the precise definition of the spatio-temporal window of its expression [73].

Then, BMPs exert pleiotropic effects during CNS pre- and post-natal development, with their phenotypic impact depending on a combination of factors, including a precise combination of ligands and receptors, the expression of other morphogens, and the epigenetic state of target cells.

## 3. Beyond Development: The Well-Known Complex Role of BMP in Cancers

The notion that cancers share common features with development, and that developmental processes may be hijacked or reactivated in tumors, emerged in the 90s; accordingly, if the activity of the BMP pathway is precisely regulated during development, alterations of this signaling cascade have been associated with tumor initiation and escape [63,64]. BMPs’ role is, however, complex in cancers. Indeed, the TGF-β super family is a prototypical example of cues showing both oncogenic and tumor suppressor functional duality, depending on the tumoral context.

Oncogenic functions of BMPs have been supported by the identification of ligand gain in several malignancies such as hepatocellular carcinoma or melanoma [74,75,76]. The oncogenic potential of the BMP pathway has been correlated with its ability to promote metastatic dissemination by enhancing tumor cells migratory and invasive properties [75,76,77]. Indeed, activation of the canonical pathway triggers the expression of matrix metalloproteinases (MMPs), leading to the remodeling of the extracellular matrix at the tumor niche and increasing the ability of tumor cells to invade surrounding tissues [78]. However, this pro-invasive effect is undoubtedly context-dependent since the opposite effect is observed in certain cancers, such as cholangiocarcinoma, or during the formation of breast cancer bone metastases [79,80,81,82,83]. In the same way, opposite roles of BMP have been described regarding tumor growth, with a tumor suppressor role attributed to a decrease in tumor cell proliferation, the self-renewal capacities of cancer stem cells, or tumoral angiogenesis [84,85], while a pro-oncogenic role has been attributed to the promotion of a quiescent state, enabling treatment resistance and subsequent regrowth post-treatment [86,87].

This paradoxical role of BMPs in cancer is illustrated extremely well by their impact on adult gliomas. First, BMPs—particularly BMP2, BMP4, and BMP7—are highly expressed in glioma tumors, suggesting a rather pro-oncogenic role. Yet, counter-intuitively, several studies have shown that the treatment of glioma cells with BMPs leads to the reduction of their proliferation rate in vitro and in murine xenograft models, with this effect being attributed to a pro-differentiating effect and/or the promotion of quiescent state [88,89,90,91,92,93]. BMP signals have notably been associated with a decrease in the expression of the stemness marker CD133 and an increase in the astrocytic GFAP marker, together with a decrease in the clonogenic properties of GSCs [89,91]. Similarly, treatment of glioblastoma stem-like cells with a mutant form of BMP7 with enhanced activity leads to a decrease in stem cell markers expression, such as SOX2, Nestin or Nanog [93]. Interestingly, treatment of these cells with BMP7 induces both GFAP and neuronal marker β-tubulin class III (TUBB3) levels, reflecting the pro-differentiating neuron/astrocyte impact of BMP during CNS development. BMP7 is also expressed by healthy neural precursor cells from the SVZ and accumulates at the tumor border, leading to a similar paracrine anti-proliferative and pro-differentiating effect [88]. In any case, and independently of the differentiation trajectory of the GSCs, reducing cell proliferation is one of the challenges of anti-cancer therapies. BMP signaling activation could also sustain aggressive tumor properties, notably by promoting invasion, as exemplified by the reduced propensity of *BMPR1* KO cells to disseminate in xenograft experiments [94]. During development, the effects of BMP in gliomagenesis are multiple and largely dependent on the overall tumor context, complicating the implementation of therapeutic approaches based on their direct targeting.

## 4. When Development Meets Tumorigenesis: Emerging Roles of BMP Pathway in pDMG

A hypothesis of the role of BMP pathway activation in DIPG emerged with the identification of *ACVR1* mutations, encoding the BMP receptor ALK2, in around 20% of tumors. Other rarer cases of genetic alterations in the BMP pathway have been observed in pDMG, with missense or truncating mutations for BMP3, BMP2K, SMAD7, BMP1, BMP2, BMP8A [3]. In 85% of the cases, *ACVR1* mutation occurs in H3.1K27M mutant tumors [95,96,97]. Constitutive activation of ALK2 resulting from this mutation was shown to synergize with H3.1K27M-mediated epigenetic remodeling, in vitro and in vivo, to promote DIPG pathogenesis [98,99]. Indeed, the invalidation of the expression of *ACVR1* by genetic engineering was notably shown to reduce clonogenicity and cell proliferation, and even to trigger cell death in H3.1 and H3.3 K27M patient-derived cell lines [98,99].

A hypothesis of the general role of BMP pathway activation has been considered in mutant non-*ACVR1* tumors [100]. Initial data, notably from the comparison of cell lines pre- and post-treatment, have led to the suggestion that the overexpression of BMP ligands may have a tumor-suppressive effect by inducing tumor cells to enter quiescence. Inhibition of tumor cell proliferation by BMPs was shown to depend on a transcriptional switch mediated by the p21-Rb-E2F axis and is highly dependent on H3.3K27M status [100,101].

This clear-cut view has changed, however, with the recent demonstration by two independent studies of the oncogenic impact of BMPs, and, in particular, of the role of BMP2 and BMP7, on the invasion potential of tumor cells. Using 3D glioma stem cell models derived from patient tumors, Bruschi et al. [102] have recently demonstrated a positive correlation between the invasive capacity of pDMG tumor cells and the autocrine production of BMP7. Complementarily, by combining the integration of single-cell data, spatial transcriptomics, and the use of patient-derived cell line spheroids, we have shown that the invasive state indeed correlates with the level of activation of the BMP pathway in DIPG models [101]. This activation could result from a complex and dynamic combination of tumor–autonomous–constitutive or stress-induced-, and of microenvironment-driven mechanisms. Indeed, the spatio-temporal window of pDMG occurrence correlates with an increased expression of BMP ligands and, notably, of BMP7 in the brain, which could create a first oncogenic initiating signal [98,100] (Figure 2). In addition to this signal from the microenvironment, tumor–autocrine production could be assumed, being induced for example by hypoxia stress [103]. The phenotypic complexity associated with the activation of the BMP pathway could be summed up in this way in pDMGs, with the establishment of a quiescent but invasive state, conferring on this pathway a role whose oncogenic status, rather than being exclusively tumor-suppressive, remains to be clarified as a function of tumor progression.

What does seem clear, however, is the interdependence between the effects of BMP pathway activation and K27M-induced epigenetic remodeling. This interdependence could be expressed from the earliest stages of tumor initiation, and condition oncogenic transformation. Indeed, the elegant study by Jessa et al. supports the view that H3.1K27M and H3.3K27M pDMG tumors likely arise from two different populations of OPC, emerging, respectively, from the ventral and dorsal zones of the neural tube. While BMP ligands are highly expressed in the dorsal part of the neural tube during development, the ventral zone concentrates the Shh morphogen. Then, the oncogenic transformation of ventrally emerging mutant H3.1 tumors would depend on the acquisition of an oncogenic mutation in *ACVR1*, explaining the strong association between these two genetic alterations. On the other hand, priming dorsally positioned H3.3K27M cells by BMP ligands, and particularly BMP7, present in the microenvironment, would be sufficient to set up a regulatory loop leading to an autocrine BMP production by the tumor, according to an amplification mechanism already described during development [99].

The triptych between epigenetic permissiveness due to histone H3 mutations, the local availability of BMPs linked to the developmental context or the acquisition of additional mutations, and the transcriptional switch induced by the BMP pathway would, therefore, condition in a major way the process of initiation and tumor escape of pDMG cells towards cellular states with oncogenic potential.

## 5. What Is Next? Therapeutic Potential of Targeting the BMP Pathway in pDMG

The process of transformation and tumor escape in pDMG points to the key role of the oncogenic synergy between chromatin remodeling induced by histone H3 mutation and the activation of the BMP pathway, although this BMP-dependent mechanism is probably not exclusive, given the heterogeneity of BMP-activation levels observed in patients’ tumors [101]. Recent evidence from the characterization of a panel of models and transcriptomic data suggests that targeting this crosstalk could be a relevant therapeutic avenue for blocking tumor cell dissemination and slowing disease progression [101,102].

Direct targeting of the BMP pathway has already been undertaken in the clinic, notably through a Phase I trial studying the impact of BMP4 administration by CED in adult glioblastoma patients [104,105]. However, considering the pleiotropic role of BMPs and their physiological importance, including during post-natal development, their use in the treatment of children and adolescents may lead to significant toxicity. pDMGs appear to exploit, hijack, and reactivate developmental processes involving the BMP pathway: a comparative approach with the mechanisms involved in CNS regulation during development will, therefore, potentially (i) provide a better understanding of the phenotypic impact linked to the activation of the BMP pathway and (ii) identify the signaling nodes conditioning the BMP/H3K27M synergy, likely to constitute the Achilles’ heels of the tumor. Indeed, it appears wiser to target the BMP pathway’s oncogenic target genes specifically activated in a tumoral context to avoid side-effects associated with the invalidation of the upstream BMP pathway, in a global and non-specific manner. The idea here would be to precisely define the pathways and effectors induced by the activation of the BMP pathway in pDMGs that are notably sufficient and necessary for the acquisition and maintenance of the invasive state.

## 6. Conclusions

The altered H3K27 epigenetic remodeling observed in pDMGs creates a permissive context, allowing transcriptional signaling pathways, normally involved in embryonic development, to acquire oncogenic valence. Characterizing the phenotypic consequences and molecular mechanisms responsible for the effects observed could help identify new therapeutic levers. Along this line, a comparative pDMG/CNS development approach based on the combined use of single-cell data and innovative organoid-like models will clearly be decisive in designing new therapeutic combinations targeting both the BMP-dependent invasive/quiescent cell state and the proliferative one, thereby establishing a streamlined combinatorial therapeutic approach of choice in the management of pDMG.

## Figures and Tables

**Figure 1 ijms-25-03361-f001:**
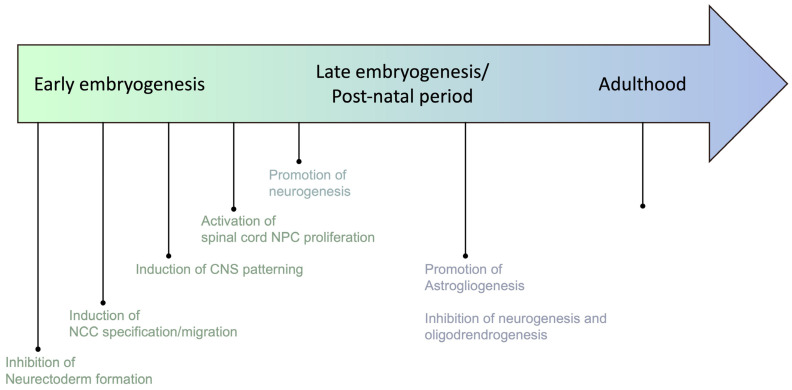
Multiple and changing roles of BMPs in the central nervous system (CNS) during development, and in adults. NCC: Neural Crest Cells. NPC: Neural Progenitor Cells. NSC: Neural Stem Cells.

**Figure 2 ijms-25-03361-f002:**
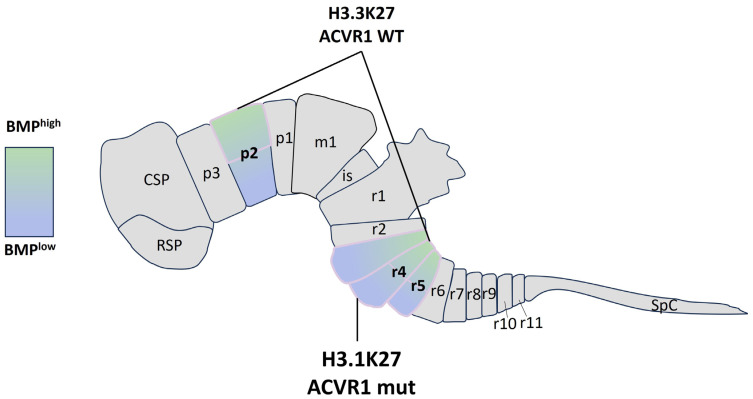
Correlation between the preferred sites of appearance of (H3.3K27M) *ACVR1* wt and (H3.1K27M) *ACVR1* mutant tumors in the territories of putative emergence of the cells of origin of DMGs. CSP: Caudal Secondary Prosencephalon; RSP: Rostral Secondary Prosencephalon; p2: Prosomere 2; m1: mesomere 1; is: isthmus; R3–4–5: Rhombomeres; SpC: developing spinal cord.

**Table 1 ijms-25-03361-t001:** Classification of pediatric gliomas according to 2021 WHO Classification of Tumors of the Central Nervous System [21].

Tumor Type	Gene/Molecular Alterations
**Pediatric-type diffuse low-grade gliomas**	
Diffuse astrocytoma	*MYB*, *MYBL1*
Angiocentric glioma	*MYB*
Polymorphous low-grade neuroepithelial tumor	*BRAF*, FGFR family
Diffuse low-grade glioma	MAPK pathway-altered *(FGFR1*, *BRAF)*
**Pediatric-type diffuse high-grade gliomas**	
Diffuse midline glioma,	H3 K27-altered, *TP53*, *ACVR1*, *PDGFRA*, *EGFR*, *EZHIP*
Diffuse hemispheric glioma,	H3 G34-mutant, *TP53*, *ATRX*
Diffuse pediatric-type high-grade glioma,	H3-wildtype and IDH-wildtype, *PDGFRA*, *MYCN*, *EGFR*
Infant-type hemispheric glioma	NTRK family, *ALK*, *ROS*, *MET*

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
