# Peer review of "Bone Morphogenic Proteins in Pediatric Diffuse Midline Gliomas: How to Make New Out of Old?"

_ijms, 2024, doi:10.3390/ijms25063361_

Round 1

Reviewer 1 Report

Comments and Suggestions for Authors

In this article, the authors reviewed about current knowledges of the oncogenic/therapeutic roles of bone morphogenic proteins (BMPs) in pediatric diffuse midline gliomas (pDMGs).

# Comments:

1. From this introduction, the motivation of the necessity of this review paper about the oncogenic/therapeutic roles of BMPs in pDMGs seemed very weak; the authors should re-write the introduction section and improve this point.

2. The authors should summarize the known basic molecular information of pDMGs by the tables and the figures, especially considering the differences with common malignant gliomas.

3. The authors should summarize the known basic information of BMP families by the tables and the figures, including their molecular signaling and the roles in normal organs.

4. The authors should summarize the current detailed knowledges of BMP-targeted molecular therapy against pDMGs by the tables and the figures.

5. The authors must add “Conclusion” section as the last section of this paper summarizing this review paper.

Author Response

Reviewer 1
# Comments:
1. From this introduction, the motivation of the necessity of this review paper about the oncogenic/therapeutic roles of BMPs in pDMGs seemed very weak; the authors should rewrite the introduction section and improve this point.

We respect the reviewer's vision, but don't share it. The introduction is there to establish a rapid state of the art on pDMGs, mentioning one of the future challenges of research on these cancers, linked to characterizing the crosstalk between epigenetic and transcriptional abnormalities, and we have added a sentence to strengthen this point. The aim is to introduce the fact that we will approach this question by focusing on the H3K27M/BMP dialogue, the
reasons for this focus being developed in the review itself.

2. The authors should summarize the known basic molecular information of pDMGs by the tables and the figures, especially considering the differences with common malignant gliomas.
We have now included a Table (Table 1) summarizing the main types of pediatric gliomas with their molecular characteristics, from the 2021 WHO classification on Central Nervous System Tumors.

3. The authors should summarize the known basic information of BMP families by the tables and the figures, including their molecular signaling and the roles in normal organs.

The functions and expression patterns of BMPs during development, ontogeny or in tissue homeostasis are multiple and dynamic in time and space: to summarize them in a table/figure is illusory. What's more, the aim of this review is not to draw up an exhaustive picture of the roles of BMPs in a physiological context, but to focus on their implications in regulating the development of the central nervous system. We have added a Figure to give an overview of
the main and dynamic roles of BMP family in nervous system development (Figure 1).

4. The authors should summarize the current detailed knowledges of BMP-targeted molecular therapy against pDMGs by the tables and the figures.

To our knowledge, and since BMP’s role starts to emerge in pDMGs, there are no BMP targeted therapy under clinical trials in pDMGs. Treatments in trial are mostly focused on ONC201 combined with chemotherapy, as mentioned in the Introduction and in the section

5. The authors must add “Conclusion” section as the last section of this paper summarizing this review paper.
This has been done.

Reviewer 2 Report

Comments and Suggestions for Authors

The epigenetic remodeling associated with these genetic alterations has also been precisely described, creating a permissive context for oncogenic transcriptional programs activation.

This review aims to describe the new findings about the involvement of BMP pathway activation in these tumors, placing their appearance in a developmental context. Targeting the oncogenic synergy resulting from this pathway activation in an H3K27M context could offer new therapeutic perspectives based on targeting treatment-resistant cell states. It appears informative and reasonable and deserved to be published in IJMS, MDPI. Only several points are raised:

1.     From cell biology and cancer research, Bone morphogenetic protein (BMP) is also known transforming growth factor B (TGF-b). The BMP pathway is one of the major signaling pathways in embryonic development, ontogeny and homeostasis, identified many years ago by pioneers in developmental biology. Although described in lines 77-93 to tell the better add more TGF-b references to provide more information of TGF-b and BMP

2.     Does new guideline (The 2021 WHO Classification of Tumors of the Central Nervous System: a summary Neuro Oncol. 2021 Aug; 23(8): 1231–1251.) involve diffuse midline gliomas (DMG) of histone H3 mutations? If so, please discuss it and add as a reference

Author Response

Reviewer 2
The epigenetic remodeling associated with these genetic alterations has also been precisely described, creating a permissive context for oncogenic transcriptional programs activation. This review aims to describe the new findings about the involvement of BMP pathway activation in these tumors, placing their appearance in a developmental context. Targeting the oncogenic synergy resulting from this pathway activation in an H3K27M context could offer new therapeutic perspectives based on targeting treatment-resistant cell states. It appears informative and reasonable and deserved to be published in IJMS, MDPI.

We thank the reviewers for his/her supportive comments.

Only several points are raised:
1. From cell biology and cancer research, Bone morphogenetic protein (BMP) is also known as transforming growth factor B (TGF-b). The BMP pathway is one of the major signaling pathways in embryonic development, ontogeny and homeostasis, identified many years ago by pioneers in developmental biology. Although described in lines 77- 93 to tell the better add more TGF-b references to provide more information of TGF-b and BMP.
We have now added two major reviews to provide more information on TGF-b and BMP.

2. Does new guideline (The 2021 WHO Classification of Tumors of the Central Nervous System: a summary Neuro Oncol. 2021 Aug; 23(8): 1231–1251) involve diffuse midline gliomas (DMG) of histone H3 mutations? If so, please discuss it and add as a reference.
Indeed, the pDMG refers to the WHO Classification. We had the Table 1 to precise this point, as recommended.

Round 2

Reviewer 1 Report

Comments and Suggestions for Authors

The authors responded to all my requests properly.